# Integrative Metabolomic and Transcriptomic Analysis Reveals the Mechanism of Specific Color Formation in *Phoebe zhennan* Heartwood

**DOI:** 10.3390/ijms232113569

**Published:** 2022-11-05

**Authors:** Hanbo Yang, Wenna An, Yunjie Gu, Jian Peng, Yongze Jiang, Jinwu Li, Lianghua Chen, Peng Zhu, Fang He, Fan Zhang, Jiujin Xiao, Minhao Liu, Xueqin Wan

**Affiliations:** 1Forestry Ecological Engineering in the Upper Reaches of the Yangtze River Key Laboratory of Sichuan Province, National Forestry and Grassland Administration Key Laboratory of Forest Resources Conservation and Ecological Safety on the Upper Reaches of the Yangtze River, Rainy Area of West China Plantation Ecosystem Permanent Scientific Research Base, Institute of Ecology & Forestry, Sichuan Agricultural University, Chengdu 611130, China; 2Sichuan Key Laboratory of Ecological Restoration and Conservation for Forest and Wetland, Sichuan Academy of Forestry, Chengdu 610081, China

**Keywords:** *Phoebe zhennan*, heartwood, golden thread, metabolomics, transcriptome, phenylpropanoid, flavonoids

## Abstract

Nanmu (*Phoebe zhennan*) is an extremely valuable tree plant that is the main source of famous “golden-thread nanmu” wood. The potential metabolites and gene regulation mechanisms involved in golden thread formation are poorly understood, even though the color change from sapwood to heartwood has been investigated in several tree plants. Here, five radial tissues from sapwood to heartwood were compared via integrative metabolomic and transcriptomic analysis to reveal the secondary metabolites and molecular mechanisms involved in golden thread formation. During heartwood formation, gradual starch grain loss is accompanied by the cell lumen deposition of lipids and color-related extractives. Extractives of 20 phenylpropanoids accumulated in heartwood, including cinnamic acids and derivatives, coumarin acid derivatives, and flavonoids, which were identified as being closely related to the golden thread. Phenylpropanoids co-occurring with abundant accumulated metabolites of prenol lipids, fatty acyls, steroids, and steroid derivatives may greatly contribute to the characteristics of golden thread formation. Additionally, the expression of nine genes whose products catalyze phenylpropanoid and flavonoids biosynthesis was upregulated in the transition zone, then accumulated and used to color the heartwood. The expression levels of transcription factors (e.g., MYB, bHLH, and WRKY) that act as the major regulatory factors in the synthesis and deposition of phenylpropanoid and flavonoids responsible for golden thread formation were also higher than in sapwood. Our results not only explain golden thread formation in nanmu, but also broaden current knowledge of special wood color formation mechanisms. This work provides a framework for future research focused on improving wood color.

## 1. Introduction

Wood color is one of the key factors determining the wood quality and economic value of tree plants, especially those used in the furniture, flooring, and other high-value decorative industries [1,2]. “Golden-thread nanmu” is a well-known, highly valuable timber industry product because of its attractive golden color, distinctive fragrance, and durability [3]. Nanmu (*Phoebe zhennan* S. Lee et F.N. Wei), which belongs to the *Phoebe* genus of the Lauraceae family, is highly valuable because of its attractive visible golden thread pattern and unique fragrance and it is the main source of “golden thread nanmu” [4,5]. This species has a long utilization history (before 221 BBC) related to the production of furniture, coffins, crafts, etc., and palace construction [3,4]. The high value of nanmu wood means that the raw wood can be sold for 5000–10,000 RMB/m^3^ (740–1500 USD/m^3^) in China. For ancient buried nanmu wood, the unit price for raw wood can reach 50,000 RMB/ton (8000 USD/ton) [5]. However, the formation of the golden thread pattern in nanmu wood is a very long, complicated process; most of the trees in which golden threads are produced in heartwood are more than 60 years old. Therefore, it is necessary to clarify the color-related metabolites and regulatory mechanisms in nanmu and provide a foundation for wood color control via molecular genetics.

Wood formation is a complex process that includes cell division, cell expansion, cell wall thickening, programmed cell death, and heartwood formation [6,7,8]. Heartwood is generally darker, denser, less permeable, and more valuable than the surrounding sapwood [9]. It is widely accepted that the special wood color of heartwood is due to the existence of extractives (secondary metabolites) [10]. Extractives (secondary metabolites) that confer heartwood with a special color may accumulate in tissues and/or cell lumens during the process of heartwood formation [6]. Such extractives (secondary metabolites) may promote a significant darkening of the wood [11]. Shao et al. [9] suggested that the deposition of stable pigments in heartwood (produced by oxidation and polymerization with the main chemical components involved in heartwood formation) was the main factor involved in the formation of heartwood color. The differences in the accumulation and composition of extractives (secondary metabolites) are responsible for the distinct colors of heartwood. Polyphenolic compounds, including anthocyanins, flavonoids, and flavones, are important common extractives responsible for heartwood coloration in many tree plants, such as *Taxus chinensis* and *Cunninghamia lanceolata* [9,12,13]. Phenolic compounds are synthesized in parenchyma cells and are then released and diffuse into adjacent cell walls and lumens, which gives the heartwood a unique color during heartwood formation [14]. In Larch red heartwood, the depth of coloration is significantly correlated with the content of phenolics in the heartwood [15]. A similar pattern was reported in *C. lanceolata*, in which the abundant accumulation of phenolic compounds, including coumarins and derivatives, flavonoids, and stilbenes, is closely related to darker color of heartwood [13]. The accumulation of norlignans (sequirin-C and agatharesinol) in the heartwood of *Cryptomeria japonica* is responsible for its darker color [16]. Cao et al. [12] concluded that phenylpropanoid, flavonoid, and lutelin glycosylation products are closely related to the red color of *C. lanceolata* heartwood. Previous studies on *Santalum album*, *Robinia pseudoacacia*, *Juglans regia*, and *Pinus sylvestris* have suggested that the dark purple, yellowish-brown, reddish-brown, and light reddish-brown heartwood colors of these tree plants are closely related to the accumulation of secondary metabolites (e.g., phenylpropanoids, polyphenols, and terpenoids) [17]. Therefore, the special color of heartwood is directly regulated by the quantity, type, and composition of extractives (secondary metabolites) that accumulate in heartwood [18].

Secondary metabolites (extractives) are the final products that directly reflect phenotypic and functional changes produced by plant metabolism driven by gene regulation. Secondary metabolites are synthesized and transported and ultimately accumulate in heartwood following the activation of metabolic pathways in plants through signal transduction during heartwood formation, resulting in the special coloration of heartwood [6]. In recent years, several studies have reported the molecular mechanisms of secondary metabolite synthesis during heartwood formation. Many upregulated genes related to the metabolism and biosynthesis of phenylpropanoids, terpenoids, flavonoids, and programmed cell death (PCD) have been detected in the transition zone in *J. regia*, *C. japonica*, and *T. cryptomerioide* based on microarray data, expressed sequence tags (ESTs), and comparative transcriptomics and they have been shown to regulate the synthesis of extractives [6,19,20]. In *Robinia pseudoacacia* and *J. nigra*, the content of flavonoids in the transition zone is closely related to the gene expression levels of phenylalanine ammonia-lyase (PAL), chalcone synthase (CHS), and flavanone-3-carboxylase dihydroflavonol-4-reductase [10,21]. Cao et al. [12,22] concluded that the genes involved in the biosynthesis of phenylpropanoids, flavonoids, and glycosyltransferases were upregulated in the transition zone of *C. lanceolata* and were responsible for the red coloration of heartwood by increasing the contents and deposition of phenylpropanoids and flavonoids. The genes responsible for naringenin production are upregulated in the leaves of a novel golden variety of *Populus deltoides* to regulate the synthesis of flavonoids [23]. A recent study also indicated that transcription factors (TFs) regulate the biosynthesis of secondary metabolites. For instance, several MYB, Bhlh, WRKY, and NAC family members have been determined to regulate the biosynthesis of flavonoids in the formation of red heartwood in *C. lanceolata* [22]. In *Salvia miltiorrhiza*, the WRKY and bHLH TFs are identified as candidate regulatory genes involved in tanshinone biosynthesis, which is related to the formation of red root color [24]. bHLH, MYB, and NAC TFs have been identified as auxin- and ethylene-responsive TFs that regulate heartwood formation in *T. cryptomerioides* [6].

These findings indicate that the formation of heartwood is an actively regulated and continuous stage of development in woody plants. Despite the significance of the secondary metabolites (extractives) and genes studied in the above tree plants for heartwood formation, there is less available information on the molecular mechanisms of the accumulation of color-related secondary metabolites. In this study, the secondary metabolites (extractives) involved in the formation of golden thread color in nanmu were inferred by integrating transcriptome and metabolome analyses. Candidate genes related to color-related secondary metabolite synthesis and regulation were initially screened. This project is the first comprehensive investigation of color-related extractives (secondary metabolites) and molecular regulation mechanisms in nanmu. This work provides fresh insight into the occurrence of special colored heartwood in “golden-thread nanmu” and will guide the future breeding of nanmu for wood production.

## 2. Results

### 2.1. Comparison of DAMs among Sapwood, the Transition Zone, and Heartwood

A total of 35,387 metabolites (16,762 and 18,625 by positive and negative models, respectively) were identified in sapwood (SW1 and SW2), TZ, and heartwood (HW1 and HW2) by LC–MS. Among these metabolites, 737 metabolites (489 and 248 by positive and negative models, respectively) were identified as common compounds detected in all tissues, and their KEGG pathways were assigned (Appendix A). These 737 metabolites were annotated in the public database after data preprocessing and classified into 13 superclasses, including 119 (16.15%) lipids and lipid-like molecules, 115 (15.60%) organic acids and derivatives, 103 (13.98%) benzenoids, 100 (13.57%) organoheterocyclic compounds, 58 (7.87%) organic oxygen compounds, and 52 (7.06%) phenylpropanoids and polyketides (Appendix A). The five tissues were separated into three consecutive groups according to the accumulation of metabolites through PCA (Figure 1A and Appendix AA,B). PC1 (the first principal component) and PC2 (the second principal component) explained 42.90% of the total variance according to the PCA of five tissues (30 samples). Furthermore, the results of hierarchical cluster analysis showed that the accumulation patterns of metabolites followed the radial distribution of the wood (Appendix A). To explore the golden thread color-related metabolites of nanmu, 339 DAMs (differentially accumulated metabolites) identified among SW1, SW2, TZ, HW1, and HW2 were screened under the following filtering conditions: VIP ≥ 1 and absolute log2 (FC) ≥ 1 (*p* ≤ 0.05) (Figure 1B). To provide further insight into DAMs associated with heartwood formation, the 339 DAMs were clustered into five clusters (C1–C5) through hierarchical cluster analysis with the default parameters (Figure 1C). The accumulated metabolites in C1 (133) were increased in heartwood and were defined as heartwood extractives. The accumulated metabolites in C3 (54) and C4 (60) were increased in the transition zone and then decreased in heartwood during heartwood formation and were defined as the intermediate products of extractive synthesis. We next assigned differentially accumulated metabolites into super class categories, including lipids and lipid-like molecules, benzenoids, phenylpropanoids and polyketides (Figure 1D). Then, to determine the metabolic pathway of DAMs, the pathway enrichment of five clusters of metabolites was identified through KEGG enrichment pathway analysis (Appendix A). The main significant pathways in C1 included phenylpropanoid biosynthesis, alpha-linolenic acid metabolism, and purine metabolism, etc., suggesting higher accumulation of these metabolites in heartwood than in sapwood. The main significant pathways in C3 and C4 included flavonoid biosynthesis, tryptophan metabolism, and flavone and flavanol biosynthesis, etc. The results of microscopy and chemical analyses also showed that flavonoids and lipids mainly accumulated in the transition zone and heartwood, whereas starch showed a decrease from sapwood to heartwood and was absent in heartwood (Figure 2).

### 2.2. Screening of Key Golden Thread-Related Metabolites in Nanmu Heartwood

OPLS-DA and S-plots were generated to explore the secondary metabolites closely related to golden thread formation in nanmu (Figure 3A–D). The R${}^{2}$ and Q${}^{2}$ values of possible OPLS-DA model overfitting were close to 1 according to a permutation test with 1000 iterations, indicating a better predictive ability of the OPLS-DA models used in our statistical analyses, showing that the models could be used for further variance analysis of secondary metabolites (Appendix A). Altogether, 190, 178, 208, and 194 upregulated metabolites identified in heartwood were chosen from HW2 vs. SW2, HW2 vs. SW1, HW1 vs. SW2, and HW1 vs. SW1, respectively (Figure 3E). According to the FC ≥ 2 of the metabolites between heartwood and sapwood, a total of 202, 208, 155, and 154 upregulated metabolites were identified in the heartwood through the comparisons of HW1 vs. SW1, HW1 vs. SW2, HW2 vs. SW1, and HW2 vs. SW2 (Appendix A). Finally, 126 significant upregulated (VIP ≥ 1 and FC ≥ 2) metabolites were chosen as potential metabolic markers closely related to golden thread formation in the heartwood of nanmu (Figure 3E,F, Appendix A). Based on compound identification, 21, 19, and 20 of 126 metabolites were classified as lipids and lipid-like molecules, benzenoids, phenylpropanoids, and polyketides. For example, the identified benzenoid compounds included benzene, substituted derivatives, and phenols; the lipids and lipid-like molecules included prenol lipids and fatty acyls; and the phenylpropanoids and polyketides included coumarins and derivatives, flavonoids, and isoflavonoids. For instance, the contents of 9(S)-HPODE, myristic acid, and stearidonic acid (the class of fatty acyls) in the heartwood were 4557.74, 131.88, and 86.20 times higher than those in sapwood (Appendix A).

### 2.3. Transcriptome Analysis and DEG Identification in Different Tissues

The clean reads (19,216,877–30,164,536) of the 18 RNA-seq libraries of SW1, SW2, and TZ were mapped to the reference genome of Phoebe bournei with a high mapping rate (85.84–90.75%) (Appendix A). A total of 4791 DEGs were identified among SW1, SW2, and TZ under the filtering criteria of an FC ≥ 1 and FDR ≤ 0.05 (Appendix A). The distribution of the DEGs in each group was visualized in volcano maps (Figure 4A–C). A total of 521 DEGs (321 downregulated and 200 upregulated), 3600 DEGs (1795 downregulated and 1805 upregulated), and 670 DEGs (387 downregulated and 283 upregulated) in SW1 vs. SW2, TZ vs. SW1, and TZ vs. SW2, respectively (Figure 4A–D). Venn diagram analysis showed that 3297 DEGs were identified in TZ vs. SW1 and TZ vs. SW2, among which 528 DEGs were included in the DEGs of both TZ vs. SW1 and TZ vs. SW2, which might be related to the biosynthesis of color-related metabolites in the heartwood (Figure 4D). Subsequently, the enrichment of the 528 DEGs in the three main functional groups (biological processes, cellular components, and molecular functions) was assessed via GO term enrichment analysis (Figure 4E). “Binding” was the largest group in the molecular function category, followed by “catalytic activity”. In the biological process category, the DEGs were mainly enriched in instances of “single-organism process”, “metabolic process”, and “cellular process”. Genes related to “membrane”, “membrane part”, “cell”, and “cell part” were predominant in the cellular component category. Starch and sucrose metabolism, plant–pathogen interaction, plant hormone signal transduction, and MAPK signaling pathway-plant were the dominant enriched pathways of the 528 DEGs according to the KEGG pathway analysis (Figure 4F,G). In the metabolic process category, almost all DEGs were divided into carbohydrate metabolism, global and overview maps, lipid metabolism (e.g., biosynthesis of fatty acids and unsaturated fatty acids), and metabolism of cofactors and vitamins (Appendix A). There were also some DEGs that were enriched in the biosynthesis pathways of color-related secondary metabolites (e.g., isoflavonoid biosynthesis, phenylpropanoid biosynthesis, and biosynthesis of tropane, piperidine, and pyridine alkaloids) and terpenoid and polyketide metabolism (e.g., biosynthesis of diterpenoids, terpenoids, and carotenoids) (Appendix A). Furthermore, the results of RT–qPCR showed that the expression patterns of selected genes were consistent with the transcriptome data (Appendix A), which suggested that the results of our RNA-seq analysis were accurate.

### 2.4. Correlation between Metabolites and Genes

Correlation analysis was performed between the metabolomic and transcriptomic data to further understand the mechanism of special golden thread color formation in nanmu heartwood. There were a large number of genes that showed a significant (*p* < 0.05) and strong correlation (R ≥ 0.8) with the metabolite contents of TZ vs. SW1 and TZ vs. SW2 (Figure 5A,B). These results indicated that the accumulation of metabolites in the transition zone and heartwood might be regulated directly or indirectly by their corresponding genes. Additionally, the KEGG co-enrichment analysis showed 146 and 130 of the same metabolic pathways in TZ vs. SW1 and TZ vs. SW2, respectively (Figure 5C,D). We ultimately identified 10 color-related metabolite-regulated pathways based on the above results. Among these pathways, 6, 2, 1, and 1 enriched fatty acid, phenylalanine, flavonoid, and isoflavonoid pathways, respectively, and were related to special color heartwood formation (Appendix A).

### 2.5. Phenylpropanoid and Flavonoid Biosynthesis Pathways Related to Nanmu Special Color Heartwood Formation

We identified the biosynthesis pathway of phenylpropanoids and flavonoids according to the data obtained from the comparative metabolomic and transcriptomic analyses among the sapwood, transition zone, and heartwood of nanmu (Figure 6). Figure 7 shows that there were obvious differences in the contents of secondary metabolites in the phenylpropanoid and flavonoid biosynthesis pathways depending on the radial tissues considered (i.e., SW1, SW2, TZ, HW1, and HW2). The high content of accumulated trans-cinnamate in the transition zone provided sufficient precursor compounds for the biosynthesis of phenylpropanoids and flavonoids. Overall, the contents of intermediate and derivative products of flavonoids in the transition zone and heartwood were generally higher than those in the sapwood. For instance, the synthesis of luteolin and dihydrokaempferol from naringenin in TZ, HW1, and HW2 was significantly higher than that in SW1 and SW2. Specifically, eriodictyol chalcone, myricetin, taxifolin, and quercetin accumulated at higher levels in the transition zone. In contrast, the products of dihydromyricetin, (−)-epicatechin, and leucocyanidin mainly accumulated in the sapwood, not in the transition zone and/or heartwood (Figure 6A). We also focused on the expression levels of genes that contributed to the biosynthesis of phenylpropanoids and flavonoids in combination with the DAMs, which regulate the synthesis of various phenylpropanoid and flavonoid compounds in the transition zone through the up- or downregulation of genes in the phenylpropanoid and flavonoid pathways. The synthesis of (−)-epicatechin and taxifolin from leucocyanidin was associated with downregulated expression levels of ANS genes (Marker00027401.gene) in the transition zone (Figure 6B). We also found one CES gene was significantly more highly expressed in the transition zone, suggesting that it could promote the accumulation of 4-coumarate in the transition zone and heartwood (Figure 6). Specifically, coniferyl aldehyde, 5-hydroxyconiferaldehyde, and sinapoyl aldehyde all accumulated in sapwood, the transition zone, and heartwood, but the contents of these metabolites in the transition zone were higher than those in sapwood.

### 2.6. Transcription Factors Related to Phenylpropanoid and Flavonoid Biosynthesis

The phenylpropanoid and flavonoid biosynthesis pathway is largely regulated by TFs, and differentially expressed TFs were identified by comparing SW1, SW2, and TZ. Thus, 252 TFs belonging to 51 TF families (bHLH, bZIP, MYB, NAC, TUB, WRKY, TUB, SBP, OFP, PLATZ, NF-X1/YA/YB/YC, EIL, etc.) were identified and used to conduct correlation analyses with genes and metabolites. As a result, 40 and 61 strongly and significantly correlated pairs (r ≥ 0.80 and *p*< 0.05) of TFs and metabolites were identified in the phenylpropanoid (Figure 7A, Appendix A) and flavonoid (Figure 7B, Appendix A) pathways, respectively. They belonged to 22 and 31 TF families consisting of bHLH, TUB, WRKY, MYB, and NAC members and played key roles in the regulation of phenylpropanoid and flavonoid pigment synthesis in nanmu heartwood formation (Appendix A). The identified TFs related to phenylpropanoids in nanmu, such as MYB and bHLH members, accounted for 12.50% (5/40) of all TFs, and those related to flavonoids accounted for 8.06% (5/62) (Figure 7A,B, Appendix A). Specifically, all five bHLH TFs (Marker00001762.gene, Marker00029745.gene, Phoebe_bournei_newGene_17495, Phoebe_bournei_newGene_7553, and Phoebe_bournei_newGene_9819) were found to be downregulated in the transition zone, which showed a strong significant negative correlation with the contents of taxifolin and myricetin, but a significant strong positive correlation with the content of leucocyanidin, indicating negative regulatory effects on the contents of taxifolin and myricetin in the flavonoid biosynthesis pathway (Figure 7B). Additionally, four MYB TFs (Marker00014790.gene, Marker00035420.gene, Marker00052819.gene, and Marker00056383.gene) were found to be significantly upregulated in the transition zone and showed a significant strong positive correlation with the contents of leucocyanidin, luteolin, quercetin, hesperetin, taxifolin, and myricetin. Nevertheless, one MYB (Marker00028131.gene) was found to be downregulated in the transition zone and showed a negative or no correlation with eight metabolites involved in flavonoid biosynthesis (Figure 7B, Appendix A). five MYB TFs (Marker00014790.gene, Marker00028131.gene, Marker00033433.gene, Marker00052819.gene, and Phoebe_bournei_newGene_13264) and five bHLH (Marker00010232.gene, Phoebe_bournei_newGene_15277, Marker00001762.gene, Marker00012962.gene, and Marker00056077.gene) showed significant strong correlations with the contents of spermidine, protocatechuic acid, genistein, and salicylic acid. Similar to flavonoid synthesis, the MYB-encoding gene (Marker00028131.gene) was also downregulated in the transition zone and showed a significant negative correlation with the contents of protocatechuic acid and genistein. Another MYB TF (Phoebe_bournei_newGene_13264) was also downregulated in the transition zone and showed a significant negative correlation with the contents of salicylic acid and protocatechuic acid. Most bHLH TFs (4/5) showed a significant negative correlation with the contents of spermidine, biochanin A, protocatechuic acid, salicylic acid, and 2-pyrocatechuic acid, and only one bHLH TF (Phoebe_bournei_newGene_15277) showed a significant positive correlation with the content of spermidine and was upregulated in the transition zone.

## 3. Discussion

In the process of this study, the differential accumulated metabolites (DAMs), physiological, and differential expression genes (DEGs) in the sapwood, transition zone, and heartwood of xylem of *Phoebe zhennan* (nanmu) were investigated and researched, respectively. A total of 737 metabolites, classified into lipids and lipid-like molecules, organic acids and derivatives, benzenoids, and phenylpropanoids and polyketides were detected in all samples. Additionally, significant tissue specificity among sapwood, transition zone, and heartwood tissues distributed in the radial direction of wood trunk was detected, which indicated that the color-related metabolites of nanmu would be transformed and synthesized in the transition zone and accumulate in the heartwood. Similar results have been reported in *Cunninghamia lanceolata*, in which the most abundant metabolites were lipids, organic acids, benzenoids, and organoheterocyclic compounds [13]. The secondary metabolites mainly accumulated in heartwood, and the primary metabolites mainly accumulated in sapwood, which provides evidence that the physiological activity and living cells of sapwood provide precursors for the synthesis of secondary metabolites through primary metabolism, after which these metabolites accumulate in heartwood during the periodic conversion of sapwood to heartwood. Similar results have been reported in many tree plants with heartwood (e.g., *C. lanceolata*, *Fagus sylvatica*, and *Taxus chinensis*) [9,13,25]. Three models (Type-I: *Robinia*, Type-II: *Juglans*, and Type-III: *Santalum*) of heartwood formation have previously been described by Celedon et al. [17] according to the mode of secondary metabolite biosynthesis and accumulation. In this study, we speculated that the model of heartwood formation in nanmu could be classified as Type-II (*Juglans*) and that extractive (secondary metabolite) biosynthesis in the transition zone occurred using precursors produced by the metabolism of starch in parenchyma cells in sapwood [14] according to the data of metabolomics, physiological, and RNA yield. These color-related secondary metabolites are diffused and deposited in the lumen and walls of nearby cells and then confer the heartwood with its special golden-thread color. The deposition of stable pigments (secondary metabolites) is the main factor responsible for wood color formation [9,26]. Phenolic compounds are commonly identified metabolites in heartwood, and flavonoids were the first rare proanthocyanidins extracted from nature. Many tree plants contain multiple types of flavonoid metabolites in their heartwood [17,27]. Previous reports have suggested that flavonoids are important common secondary metabolites related to color in plants [9,12]. In Morus (e.g., *Morus tinctoria*, *M. alba*, and *M. bambycis*), the heartwood is known as old fustic due to the accumulation of morin (a bioactive flavonoid compound) [27]. In *C. lanceolata*, the accumulation of flavonoids, phenylpropanoids, coumarins, and derivatives thereof in heartwood is closely related to the red color of the heartwood [13,22]. Benzenoids-phenylpropanoids are the main secondary metabolites responsible for the flower color and aroma of *Camellia sinensis* [28]. The accumulation of phenylpropanoids, phenolics, and terpenes causes the heartwood of *Santalum album*, *Robinia pseudoacacia*, *Juglans regia*, and *Pinus sylvestris* to become dark purple, reddish-brown, red, or chestnut red-brown [29]. In our study, the amount of secondary metabolites (benzene, phenols, prenol lipids, fatty acyls, coumarins, flavonoids, and isoflavonoids, etc.) and their conjugated structures confer an attractive golden yellow color and aromatic odor and are considered the main contributors to the unique color and fragrance of the heartwood of nanmu. The abundant accumulated metabolites of lipids and lipid-like molecules (e.g., linoleic acid, 9(S)-HPODE, myristic acid, and stearidonic acid) that co-occur with these metabolites may greatly contribute to the characteristics of golden thread formation in nanmu wood. Therefore, we boldly speculated that the lipids (steroids and steroid derivatives, prenol lipids, fatty acyls, etc.) and phenylpropanoid metabolites (cinnamic acids and derivatives, flavonoids, isoflavonoids, linear 1,3-diarylpropanoids, and phenylpropanoic acids) that accumulate in heartwood are the main extractives responsible for golden thread formation in nanmu. Furthermore, the accumulated organic acids and derivatives are related to the durability of wood.

The results of RNA-seq analysis confirmed the results of the DAMs analysis and demonstrated that the primary metabolites in sapwood provide precursors for color-related secondary metabolite biosynthesis (biosynthesis of fatty acids and unsaturated fatty acids, phenylpropanoid biosynthesis, flavonoid and isoflavonoid biosynthesis, biosynthesis of tropane and piperidine) in the transition zone. Broadly similar results have recently been obtained in *T. cryptomerioides*, *J. nigra*, and *R. pseudoacacia*, and the expression levels of phenylpropanoid and flavonoid biosynthesis genes were shown to be closely related to the contents of these secondary metabolites in the transition zone and heartwood [6,10,30]. Insights into the biosynthesis pathways of the key metabolite compounds are conducive to clarifying the evolution of metabolites and they provide a foundation for achieving the selective and/or increased production of target metabolites [31]. ANS (anthocyanidin synthase) catalyzes the reaction transforming leucoanthocyanidin (colorless) into anthocyanidin (colored) [32]. Here, we observed that the synthesis of (−)-epicatechin and taxifolin from leucocyanidin was associated with downregulated expression levels of ANS genes in the transition zone. This showed that the pathways of flavonoid metabolites were not enriched in the colored anthocyanidin biosynthesis pathway during nanmu heartwood formation. However, in *C. lanceolata*, glycosylated flavonoids provide precursors for anthocyanidin synthesis and are the key metabolites responsible for red heartwood [12]. These compounds are responsible for large differences in the color of heartwood, although they share a large number of secondary metabolites that accumulate in heartwood. CHI (chalcone isomerase) is the key enzyme in flavonoid biosynthesis and participates in the regulation of flavonoid compound synthesis in tree plants (e.g., *C. lanceolata* and *Hopea hainanensis*) [12,31]. The overexpression of CHI increased the levels of flavonoids [33]. In our study, CHI was found to be upregulated in the transition zone, which was accompanied by the upregulation of its downstream counterpart naringenin. This indicated the accumulation of precursors of color components in the golden thread of nanmu heartwood. In addition, phloretin contents significantly increased from SW1 to HW2, with the up- and/or downregulated expression of PGT1 (phlorizin synthase) genes in the transition zone. Downstream genes such as COMT and F5H would be recruited, and their expression levels would be promoted by the accumulation of these upstream metabolites, which could further drive the production of downstream metabolites (e.g., caffeate, ferulate, 5-hydroxyferulic, and sinapate) as precursors for soluble and cell wall-bound phenolic generation [34]. Specifically, coniferyl aldehyde, 5-hydroxyconiferaldehyde, and sinapoyl aldehyde all accumulated in sapwood, the transition zone, and heartwood, but the contents of these metabolites in the transition zone were higher than those in sapwood. In contrast, the expression of the genes (CCR and CAD) responsible for the synthesis of these metabolites was downregulated in the transition zone, which could negatively regulate the biosynthesis of sinapoyl aldehyde and sinapyl alcohol, the precursors of syringyl lignin subunit formation [34]. Therefore, we speculated that metabolites in the phenylpropanoid pathway were more likely to be channeled toward soluble and cell wall-bound phenolic production than syringyl lignin subunit generation in nanmu heartwood formation. It is widely accepted that phenylpropanoid biosynthesis is the upstream pathway of the flavonoid biosynthesis [22]. In this study, a number of DAMs and DEGs in the phenylpropanoid biosynthesis pathway were identified; nevertheless, downstream of the phenylpropanoid pathway, relatively few DAMs and DEGs were identified in the flavonoid biosynthesis pathway. The phenylpropanoid biosynthesis pathway is not restricted to the biosynthesis of flavonoids and lignans, but also contributes to the production of other aromatic metabolites, such as phenolic volatiles and coumarins [35]. Therefore, the phenylpropanoid pathway was identified as the main metabolite biosynthesis pathway in nanmu, in which synthases, accumulated aromatic phenylpropanoids, and intermediate and downstream products in the transition zone and heartwood, such as flavonoids, coumarins and derivatives, cinnamic acids and derivatives thereof, and phenylproanoic acids, caused the heartwood to develop its special golden thread color and fragrance. We also identified some differentially expressed genes related to beta-glucosidase and peroxidase production in the phenylpropanoid biosynthesis pathway between sapwood and the transition zone, showing positive or negative regulation mechanisms in the biosynthesis of phenylpropanoids.

TFs are involved in the regulation of plant growth and development, including secondary metabolite biosynthesis, by acting as trans-acting factors by activating or inhibiting the expression of genes [31,36,37]. Recent studies have demonstrated that the expression levels of genes related to flavonoid biosynthesis are often regulated by the interactions of TF families [31]. MYB and bHLH TFs are the two largest TF families in plants that regulate the biosynthesis of flavonoid pigments [31,38,39]. Cao et al. [22] identified three MYB TFs that jointly modulate flavonoid biosynthesis in *C. lanceolata* red heartwood formation. In our study, as MYB and bHLH genes are related to the accumulation of flavonoid compounds, it was expected that these TFs would be significantly up and/or downregulated in the transition zone, and they showed a strong significant correlation with the contents of flavonoid metabolites in nanmu. In this study, five MYB and five bHLH TFs showed strong significant correlations with the contents of spermidine, protocatechuic acid, genistein, and salicylic acid. Most bHLH TFs showed a significant negative correlation with the contents of spermidine, biochanin A, protocatechuic acid, salicylic acid, and 2-pyrocatechuic acid, and only one bHLH TF showed a significant positive correlation with the content of spermidine and was upregulated in the transition zone. Therefore, increases in golden thread and fragrance pigments seemed to be driven by the positive or negative regulation of phenylpropanoid- and flavonoid-related TFs. In the next step, we will further clarify the roles of these TFs and provide a more accurate theoretical framework for the directional improvement of wood traits in nanmu.

## 4. Materials and Methods

### 4.1. Plant Materials and Sampling

Nanmu (*Phoebe zhennan*) trees were planted in the Du Fu Thatched Cottage Museum in Chengdu (N $30^{\circ }39^{\prime }37^{\prime \prime }$, E $104^{\circ }1^{\prime }41.69^{\prime \prime }$) in Sichuan Province, China. Previous studies had proved that season of heartwood formation in most wood plants ended the quiescent dormant stage during cambial reactivation and the thick walls during dormancy store material that can be metabolized in spring [40]. Additionally, the death of ray parenchyma cells at intermediate wood occurred in spring [41]. Therefore, the north–south increment core samples collected at breast height (1.3 m) in six sib-families of nanmu (80 years old) were sampled at 9:00–10:00 a.m. in April 2021 (Figure 8A). The color of heartwood was a deeper yellow color than sapwood. Thus, the increment core from each tree was divided into five tissues (SW1: outer sapwood, SW2: inner sapwood, TZ: transition zone, HW1: outer heartwood, HW2: inner heartwood) by visual observation, and six replicates of each tissue type were performed for each tree (Figure 8B). 5–7 growth rings were included in SW1, SW2, HW1, and HW2, respectively, and 2–3 growth rings were included in TZ. The SW1, SW2, TZ, HW1, and HW2 tissues of the north and south increment cores were mixed into one sample. Immediately thereafter, each increment core was divided into two parts along a cross-section in the middle of the increment core (Figure 8C). One part was stored at −80 ${}^{\circ }$C for flavonoid measurement, RNA, and metabolite extraction. Another part was fixed and subjected to vacuum treatment with 4% paraformaldehyde for subsequent immunohistochemical staining analysis.

### 4.2. Microscopy Analyses and Flavonoid Content Measurement

Radial sections (30 μm in thickness) from each tissue were obtained at −20 ${}^{\circ }$C using a cryostat (Leica CM1900). To observe the starch grains and lipids, radial sections from five tissues were subjected to I${}_{2}$-KI and Sudan IV staining. After the staining process, the sections were gradually dehydrated and cleared with 30% (5 min), 50% (5 min), 70% (5 min), 90% (5 min), 95% (5 min), and 100% alcohol (10 min), alcohol: dimethylbenzene (v:v = 1:1) (2 min) and dimethylbenzene (4 min). Then, the sections were mounted on microscope slides and observed under a microscope (Olympus BX50). SW1 and SW were mixed to form a single sample, designated SW (sapwood), and HW1 and HW2 were mixed to form a single HW (heartwood) sample. Then, the flavonoid contents of SW, TZ, and HW were measured using a plant flavonoid test kit (Nanjing Jiancheng Bioengineering Institute, Nanjing, China).

### 4.3. Identification and Quantification of Metabolites

A 100 mg sample was ground into powder using a tissue grinder (Kunshan, China) with glass beads (Sigma–Aldrich, Shanghai, China) for 90 s at 60 Hz. Metabolites were extracted at room temperature by applying ultrasound for 15 min in 600 μL methanol (Fisher Scientific, Loughborough, UK) (containing 2-amino-3-(2-chlorophenyl)-propionic acid (4 ppm)). Following centrifugation for 10 min at 4 ${}^{\circ }$C and 12,000 rpm, the supernatant was removed and syringe filtered with microporous membrane filters (0.22 pim, Tianjin Jinteng Experiment Equipment Co., Ltd., Tianjin, China) before LC–MS analysis. The metabolites were analyzed using a UHPLC system (Thermo Fisher Scientific, Waltham, MA, USA). Metabolite quantification and identification were performed by PANOMIX (Chengdu, China). Metabolites were identified using the public HMDB [42], massbank [43], LipidMaps [44], mzcloud [45], and KEGG [46] databases and the self-built database of BioNovoGene (Chengdu, Sichuan, China) (http://query.biodeep.cn/, v1.0.0.2, accessed on 25 August 2022), with the following parameters: retention time, ppm (<30 ppm), and fragmentation model. Differentially accumulated metabolites (DAMs) were screened under the following filtering conditions: variable importance in projection (VIP) ≥ 1 and absolute log2 (fold change (FC)) ≥ 1 (*p*≤ 0.05). Principal component analysis (PCA) performed with the “rplos” package in R and orthogonal partial least squares-discriminant analysis (OPLS-DA) in MetaboAnalyst 5.0 [47] were used to examine the patterns of metabolite accumulation in different tissues and the accumulation of specific metabolites.

### 4.4. RNA-Seq Analysis

Total RNA was extracted from each increment core and purified from DNA, resinous substances, etc using a plant RNA extraction kit (TaKaRa, Dalian, China) according to the manufacturer’s instructions. The purity, integrity, and concentration of the RNA was tested by using Nanodrop and 2100/GX instruments (Appendix A). The results of RNA integrity testing showed that the integrity of RNA in heartwood (HW1 and HW2) did not satisfy the requirements of transcriptome sequencing library construction (Appendix A). Therefore, the mRNA libraries of sapwood (SW1 and SW2) and the transition zone (TZ) were constructed and sequenced on the Illumina HiSeq 4000 platform, and low-quality adapters were then removed using Fastp with the default parameters [48]. Then, HISAT2 was used to align the clean reads to the reference genome (*Phoebe bournei*) [49,50]. DESeq2 software was used to determine gene/transcript expression at the gene level according to the FPKM method based on the raw count data [51]. DEGs (differentially expressed genes) were defined according to an FC ≥ 1 and an FDR (false discovery rate) ≤ 0.05 and subjected to gene ontology (GO) and Kyoto Encyclopedia of Genes and Genomes (KEGG) enrichment analyses [46].

### 4.5. Real-Time Quantitative PCR

First-strand cDNA was synthesized using a TaKaRa first-strand cDNA synthesis kit (TaKaRa, Dalian, China). We selected 21 genes involved in phenylpropanoid and flavonoid biosynthesis for real-time quantitative PCR (RT–qPCR) analysis, and the actin gene and 18S rRNA was used as the reference gene for gene expression analysis (Appendix A). RT–qPCR was performed using a TaKaRa real-time quantitative PCR kit (TaKaRa, Dalian, China) on a CFX96 Real-Time System (BIO-RAD, Hercules, CA, USA). The relative expression level of the 21 genes were calculated by the ratio = (Et)${}_{\Delta }$${}_{\mathrm{CT}}$${}_{t}$/NF [52,53]. Three technical replicates were calculated for each sample.

### 4.6. Integrative Transcriptomic and Metabolomic Analysis

The association analysis among differentially expressed TFs and metabolites of phenylpropanoids and flavonoids was performed by Pearson correlation analysis. The Pearson correlation coefficient (PCC) was calculated, and screening was conducted according to the criterion of a PCC ≥ 0.80 or ≤−0.80 (*p*< 0.05). The interaction networks between differentially expressed TFs and DAMs were mapped by using Cytoscape version 3.10.0.

## 5. Conclusions

The metabolomes and transcriptomes of the sapwood, transition zone, and heartwood of nanmu (*Phoebe zhennan*) were compared to explore the biochemical basis and regulatory mechanism of golden thread formation. The results provide insights into the lipids and lipid-like molecules, phenylpropanoids, and flavonoids that accumulate in the heartwood, which are closely related to the golden thread color in nanmu. Among these compounds, 9(S)-HPODE, involved in linoleic acid metabolism, might be chosen as a key potential marker compounds. Key enzyme-encoding genes (CHI, CES, F5H, etc.) and 40/61 TFs (MYB, bHLH, WRKY, etc.) showed strong significant correlations with the contents of intermediate compound products involved in the biosynthesis of phenylpropanoids and flavonoids, respectively. Taken together, our findings broaden the knowledge of the color-related metabolites and regulatory mechanisms involved in special color formation in heartwood and provide a theoretical foundation for the development of new cultivars of nanmu wood with high-quality golden threads in future breeding work.

## Figures and Tables

**Figure 1 ijms-23-13569-f001:**
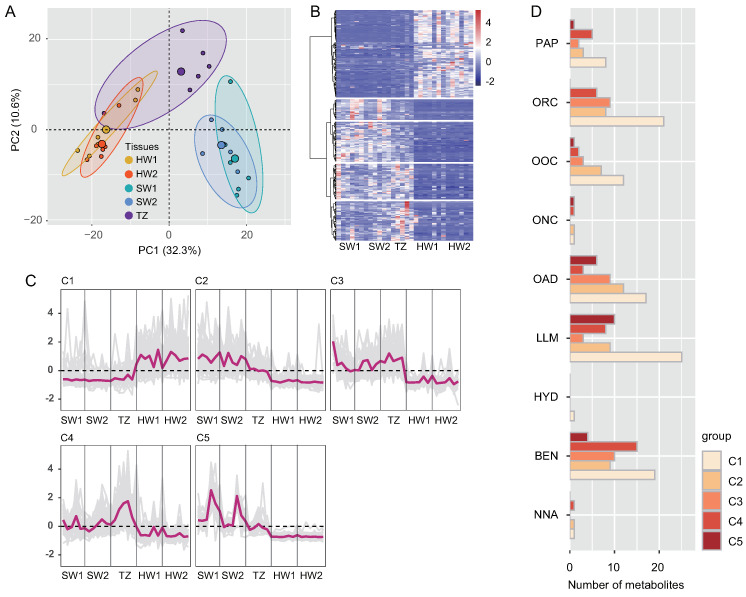
Preliminary analysis of metabolomics data. (**A**) Principal component analysis (PCA) of metabolites in five tissues. (**B**) Hierarchical clustering of the compound profiles of 339 differentially accumulated metabolites (DAMs). (**C**) Hierarchical clustering showing accumulation changes in DAMs that were categorized into five clusters (C1–C5). Auto–scaled standardization compounds in each cluster are shown. (**D**) Superclass categories in differential co–accumulated modules (C1–C5). SW1, outer sapwood, SW2, inner sapwood, TZ, transition zone, HW1, outer heartwood, HW2, inner heartwood. PAP, phenylpropanoids and polyketides, ORC, organoheterocyclic compounds, OOC, organic oxygen compounds, ONC, organic nitrogen compounds, OAD, organic acids and derivatives, LLM, lipids and lipid–like molecules, HYD, hydrocarbons, BEN, benzenoids, NNA, nucleosides, nucleotides, and analogues.

**Figure 2 ijms-23-13569-f002:**
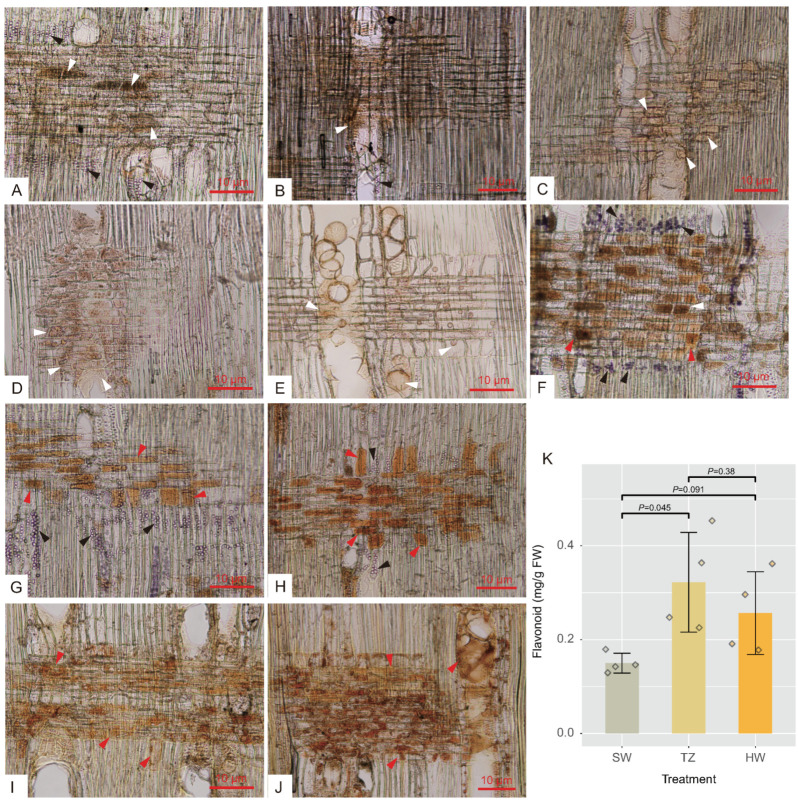
Light micrographs of starch grains, lipid, and extractives and flavonoid content along with radial sections. (**A**–**E**) Outer sapwood, inner sapwood, transition zone, outer heartwood, and inner heartwood stained with sudan IV solution showing lipid (white arrows) and starch grains (black arrows). (**F**–**J**) Outer sapwood, inner sapwood, transition zone, outer heartwood, and inner heartwood stained with I${}_{2}$-KI solution showing starch grains (black arrows) and extractives (red arrows). (**K**) Total flavonoid content of nanmu radial tissues, SW, sapwood, TZ, transition zone, HW, heartwood.

**Figure 3 ijms-23-13569-f003:**
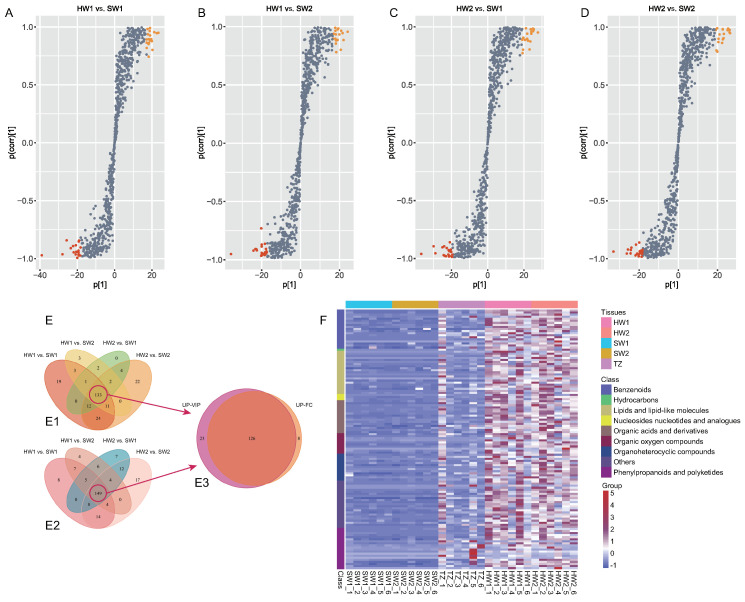
Potential metabolite markers analysis in heartwood of nanmu. (**A**–**D**) Orthogonal partial least squares-discriminant analysis (OPLS–DA) between sapwood and heartwood. The red and orange point indicated the top 20 metabolites. (**E**) Venn diagram of differentially accumulated metabolites in heartwood. E1, Venn diagram of up–accumulated metabolites in heartwood according to the results of foldchange (FC ≥ 2), E2, Venn diagram of up–accumulated metabolites in heartwood according to the results of VIP value (VIP ≥ 1), E3, Venn diagram of up−accumulated metabolites in heartwood combined with fold change and VIP value. (**F**) Cluster heatmap of the 126 DAMs significantly accumulated in the heartwood lines. SW1, outer sapwood, SW2, inner sapwood, TZ, transition zone, HW1, outer heartwood, HW2, inner heartwood.

**Figure 4 ijms-23-13569-f004:**
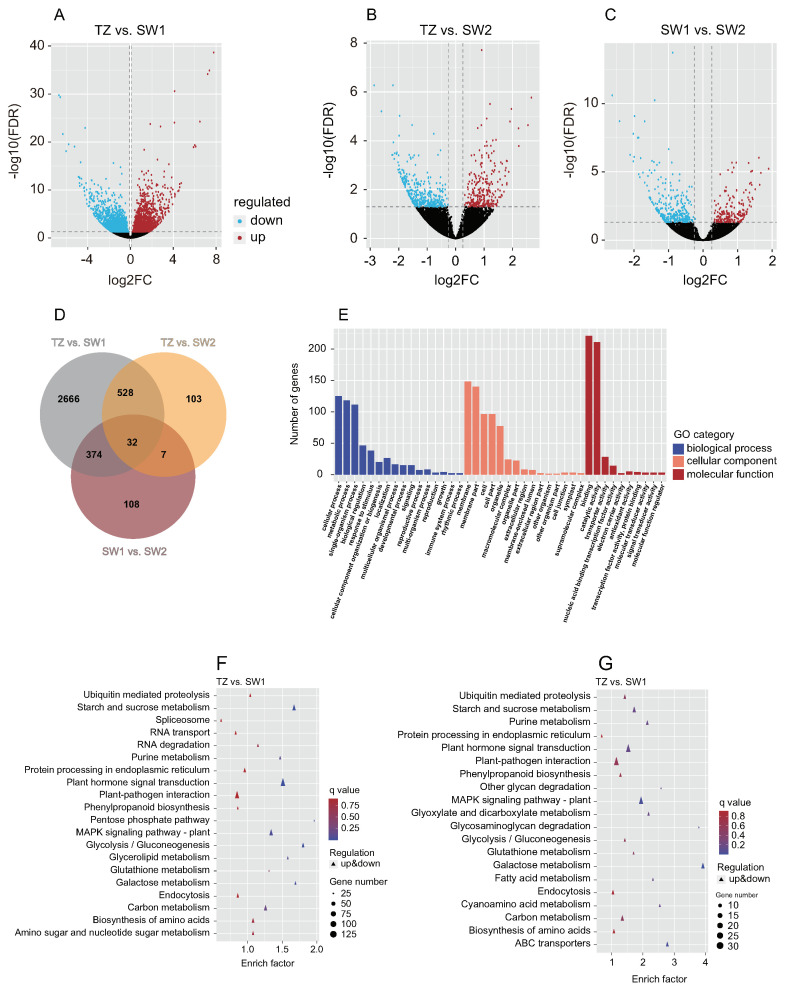
Preliminary analysis of transcriptomic data. (**A**–**C**) Volcano plots of differentially expressed genes (DEGs). (**D**) Venn diagram of DEGs. (**E**) GO enrichment analysis of DEGs in TZ vs. SW1 and TZ vs. SW2. (**F**) Top 20 KEGG pathways with the most significant enrichment in the TZ vs. SW1. (**G**) Top 20 KEGG pathways with the most significant enrichment in the TZ vs. SW2. The number of DEGs is represented by the size of circle.

**Figure 5 ijms-23-13569-f005:**
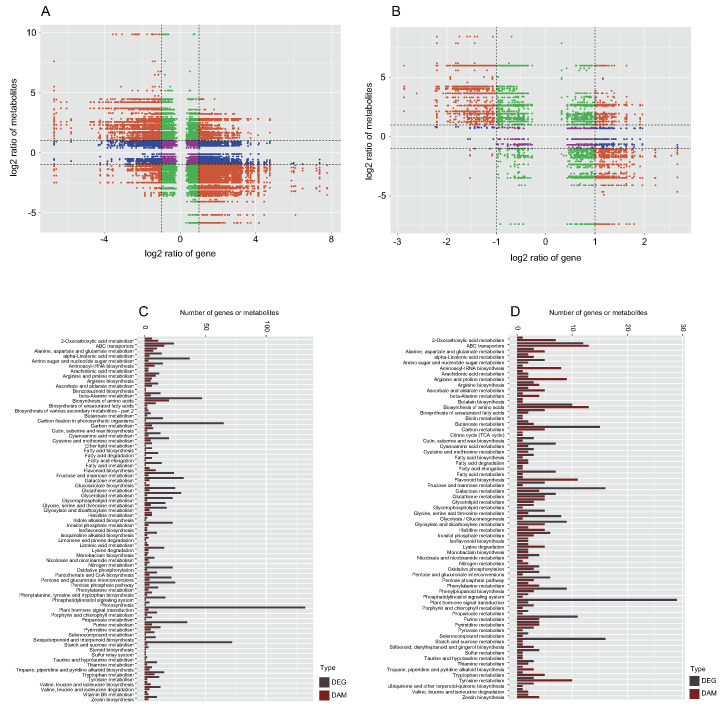
Correlation analysis of transcriptomic and metabolomic data among different tissues of *P. zhennan*. (**A**) The nine–quadrant diagram shows the correlation of genes and compounds between TZ and SW1. (**B**) The nine−quadrant diagram shows the correlation of genes and compounds between TZ and SW2. (**C**,**D**) KEGG enrichment analysis of DEGs and DAMs enriched in the same pathway.

**Figure 6 ijms-23-13569-f006:**
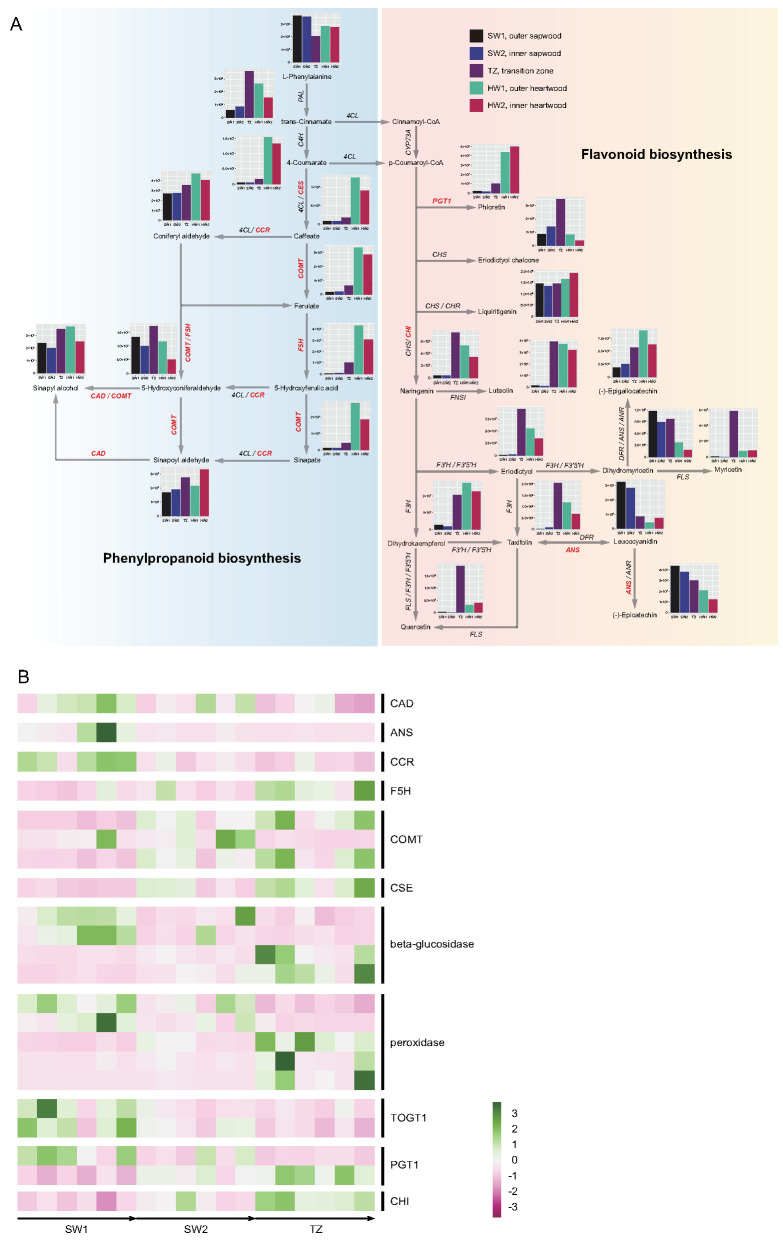
Phenylpropanoids and flavonoid biosynthesis in *P. zhennan* heartwood formation. (**A**) The biosynthesis pathway of phenylpropanoids and flavonoids. This pathway is constructed based on the KEGG pathway and literary reference. (**B**) The heatmap of differentially expressed genes among sapwood and transition zone of wood in the phenylpropanoid and flavonoid biosynthesis pathway. The histogram indicated the content of differentially accumulated metabolites in five tissues of wood. The red abbreviations indicated the differentially expressed genes among five tissues of wood. CAD, cinnamyl–alcohol dehydrogenase, ANS, anthocyanidin synthase, CCR, cinnamoyl–CoA reductase, F5H, ferulate–5–hydroxylase, COMT, caffeic acid 3–O–methyltransferase/acetylserotonin O–methyltransferase, CSE, caffeoylshikimate esterase, TOGT1, scopoletin glucosyltransferase, PGT1, phlorizin synthase, 4CL, 4–coumarate CoA ligase, C4H, cinnamate 4–hydroxylase, PAL, phenylalanine ammonia–lyase, CHS, chalcone synthase, CHI, chalcone isomerase, FLS, Flavonol synthase, F3${}^{\prime }$H, flavanone 3${}^{\prime }$–hydroxylase, F3${}^{\prime }$5${}^{\prime }$H, flavanone 3${}^{\prime }$–5${}^{\prime }$–hydroxylase, F3H, flavanone 3–hydroxylase, ANR, anthocyanidin reductase, DFR, bifunctional dihydroflavonol 4–reductase/flavanone 4–reductase, CYP73A, trans–cinnamate 4-monooxygenase, FNSI, flavone synthase I.

**Figure 7 ijms-23-13569-f007:**
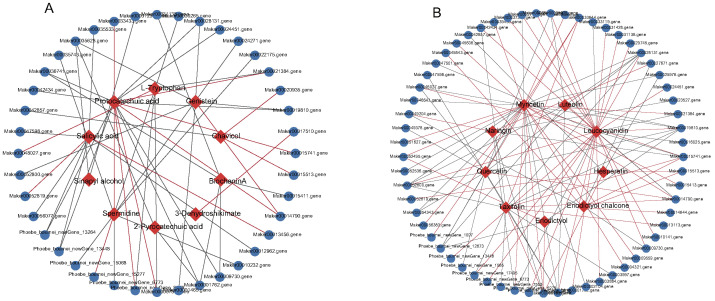
Connection network between transcript factors and phenylpropanoid, flavonoids compounds. (**A**) Phenylpropanoid and (**B**) flavonoids metabolites. The blue ellipsoid represents the DEGs, the red ellipsoid represents the DAMs, the red line indicates a positive correlation, and the grey line indicates a negative correlation.

**Figure 8 ijms-23-13569-f008:**
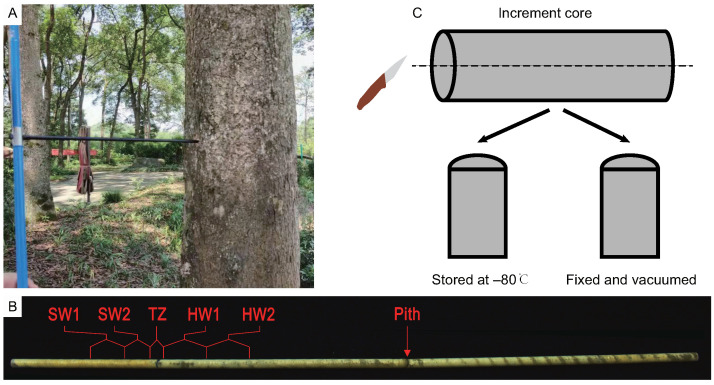
The sampling schemes of increment core of *Phoebe zhennan*. (**A**) The increment core was sampled at the breast height of trees. (**B**) The increment core was divided into five tissue parts. SW1, outer sapwood, SW2, inner sapwood, TZ, transition zone, HW1, outer heartwood, HW2, inner heartwood, the same as below. (**C**) The tissues was divided into two parts at along the cross section at the middle of increment core.

## Data Availability

The datasets presented in this study can be found in online repositories. The names of the repository/repositories and accession number(s) can be found below: PRJNA898439 (http://www.ncbi.nlm.nih.gov/bioproject/898439, accessed on 25 August 2022).

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
