# Peer review of "Integrative Metabolomic and Transcriptomic Analysis Reveals the Mechanism of Specific Color Formation in *Phoebe zhennan* Heartwood"

_ijms, 2022, doi:10.3390/ijms232113569_

Round 1

Reviewer 1 Report

The article is aimed to an important question – understanding the of «golden-thread» formation у nanmu (Phoebe zhennan), and it will be of great interest to scientific readers working with woody plants after the publication. The authors presented a large, interesting material comparing five radial tissues from sapwood to heartwood by integrating transcriptome and metabolome analyses.

Major comments:

1. The process of transition from sapwood to heartwood is extended in time. The stages of heartwood formation (starch disappearance, xylem dehydration, switching the primary metabolism to the synthesis of secondary substances, changes of the cell wall structure, nucleus disappearance) occur throughout the season. That’s why; it is premature to talk about the mechanisms of heartwood formation accordingly to the analysis of the material collected on the only one date. In my opinion, this is a weak point of work. The authors did not explain what was the reason for the choice of this date for stem tissue sampling.

2. Isolation of total RNA, qPCR.

It is not clear whether the purity of RNA from DNA, resinous substances, etc. was assessed. As a rule, special commercial kits are not adapted for working with wood samples.

Did you only use one reference gene? According to MIQE (Minimum Information for Publication of Quantitative Real-Time PCR): «Unless fully validated single reference genes are used, normalisation should be performed against multiple reference genes, chosen from a sufficient number of candidate reference genes tested from independent pathways using at least one algorithm..» (Bustin, S.A., Beaulieu, JF., Huggett, J. et al. MIQE précis: Practical implementation of minimum standard guidelines for fluorescence-based quantitative real-time PCR experiments. BMC Molecular Biol. 2010. 11, 74. https://doi.org/10.1186/1471-2199-11-74)

Minor comments:

1. Introduction, line 27. Latin name of the species should be written in italics «…Phoebe zhennan…»

2. Introduction, lines 35-36. «…are more than 60 a old.» What does it mean: 60 years old?

3. Introduction, line 60. Latin name of the species should be written in italics «…Cryptomeria japonica …»

4. Introduction, line 87. Latin name of the species should be written in italics «…Populus deltoides…»

5. Resuits, line 248. «…and flavonoid (Figure 7B, Table S10Maybe « Table S11..»?

6. Materials and Methods. Plant materials and sampling. The age of the trees is not detected.

7. Materials and Methods. Plant materials and sampling. It is not clear how heartwood and sapwood were detected on the increment core. Did you make it only visually? How many growth rings were included in each tissue (outer sapwood, inner sapwood, transition zone, outer heartwood, inner heartwood)?

8. Materials and Methods, line 423. «adial…» Maybe «Radial..»?

9. Materials and Methods, line 238. «…2-amino-amino-3-(2-chloro-phenyl)-propionic acid…» It should be written in this way: «…2-amino-3-(2-chlorophenyl)-propionic acid…»

Kind regards

Author Response

List of Responses

Dear Editors and Reviewers:

Thank you for your letter and for the reviewers’ comments concerning our manuscript entitled “Integrative metabolomic and transcriptomic analysis reveals the mechanism of specific color formation in Phoebe zhennan heartwood (ijms-1909297)”. Those comments are all valuable and very helpful for revising and improving our paper, as well as the important guiding significance to our researches. We have studied comments carefully and have made correction which we hope meet with approval. Revised portion are marked in red in the paper. The main corrections in the paper and the responds to the reviewer’s comments are as following:

Responds to the reviewer’s comments:

  1. The process of transition from sapwood to heartwood is extended in time. The stages of heartwood formation (starch disappearance, xylem dehydration, switching the primary metabolism to the synthesis of secondary substances, changes of the cell wall structure, nucleus disappearance) occur throughout the season. That’s why; it is premature to talk about the mechanisms of heartwood formation accordingly to the analysis of the material collected on the only one date. In my opinion, this is a weak point of work. The authors did not explain what was the reason for the choice of this date for stem tissue sampling.

Response to comment:

It is really true as reviewer point out that the stages of heartwood formation (starch disappearance, xylem dehydration, switching the primary metabolism to the synthesis of secondary substances, changes of the cell wall structure, nucleus disappearance) occur throughout the season. We are very sorry for our negligence of the writing of sampling schemes. Previous studies had proved that season of heartwood formation in most wood plants ended the quiescent dormant stage during cambial reactivation and the thick walls during dormancy store material that can be metabolized in spring (Kampe et al, 2013). The death of ray parenchyma cells at intermediate wood occured in spring (Nakada et al, 2012). Therefore, the reason of spring (April) of this date for stem tissue sampling in our study. The reason for the choice of this date for sampling have been supplied in the section of materials and methods. Please review line 411-415.

Reference

Kampe A. and Magel E. New insights into heartwood and heartwood formation. In: Fromm J. (eds) cellular aspects of wood formation. Plant Cell Monographs, Springer, Berlin, Heidelberg, 2013, vol 20, 71. (doi: https://doi.org/10.1007/978-3-642-36491-4_3)

Nakada R. and Fukatsu E.. Seasonal variation of heartwood formation in Larix kaempferi. Tree Physiology, 2012, 32 (12): 1497-1508.

  1. Isolation of total RNA, qPCR.

It is not clear whether the purity of RNA from DNA, resinous substances, etc. was assessed. As a rule, special commercial kits are not adapted for working with wood samples.

Response to comment:

As reviewer point out that special commercial kits would be not adapted for working with wood samples. Some commercial kits (e.g., TaKaRa, TIANGEN, Invitrogen, and Vazyme) were selected to isolated the total RNA of wood. Finally, the high-quality total RNA was isolated by one of RNA extraction kit (TaKaRa, Dalian, China). The DNA, resinous substances, etc were removed by DNase I (with 10×DNase I buffer and Recombinant DNase I), Buffer PE, and Buffer NB according to the manufacture’s instructions. The results of RNA quality test were showed in Figure S11, A-C. The procedure of the purity of RNA was supplied in the section of RNA-seq analysis in the section of materials and methods. Please review line 462-465.

Did you only use one reference gene? According to MIQE (Minimum Information for Publication of Quantitative Real-Time PCR): «Unless fully validated single reference genes are used, normalisation should be performed against multiple reference genes, chosen from a sufficient number of candidate reference genes tested from independent pathways using at least one algorithm..» (Bustin, S.A., Beaulieu, JF., Huggett, J. et al. MIQE précis: Practical implementation of minimum standard guidelines for fluorescence-based quantitative real-time PCR experiments. BMC Molecular Biol. 2010. 11, 74. https://doi.org/10.1186/1471-2199-11-74)

Response to comment:

Considering the reviewer’s suggestion, we have re-analyzed the gene expression level of Quantitative Real-Time PCR using two reference genes (Actin and 18S rRNA). The data of normalization of the two reference genes were used to calculate the relative expression levels. The relative expression level of the 20 genes were calculated by the ratio=(ET)ΔCTt/NF (Bustin et al, 2010; Wang et al, 2014). Please review line 480-484, and Table S11, Figure S10.

Reference

Bustin S.A., Beaulie J.F., Huggett J, et al. MIQE précis: Practical implementation of minimum standard guidelines for fluorescence-based quantitative real-time PCR experiments. BMC Molecular Biol. 2010. 11, 74.

Wang H.L., Chen J, Tian Q, et al. Identification and validation of reference genes for Populus euphratica gene expression analysis during abiotic stresses by quantitative real-time PCR. Physiologia Plantarum, 2014, 152, 529-545.

  1. Introduction, line 27. Latin name of the species should be written in italics «…Phoebe zhennan…»

Response to comment:

We are very sorry for our negligence, and we have made correction. Please review line 27.

  1. Introduction, lines 35-36. «…are more than 60 a old.» What does it mean: 60 years old?

Response to comment:

Yes, it mean 60 years old, and we have made correction in the text. Please review line 36.

  1. Introduction, line 60. Latin name of the species should be written in italics «…Cryptomeria japonica …»

Response to comment:

We are very sorry for our negligence, and we have made correction. Please review line 60.

  1. Introduction, line 87. Latin name of the species should be written in italics «…Populus deltoides…»

Response to comment:

We are very sorry for our negligence, and we have made correction. Please review line 87.

  1. Resuits, line 248. «…and flavonoid (Figure 7B, Table S10)» Maybe « Table S11..»?

Response to comment:

We are very sorry for our negligence, and we have made correction. Please review line 248.

  1. Materials and Methods. Plant materials and sampling. The age of the trees is not detected.

Response to comment:

Six trees (80 years old) were selected to sample in this study. We are very sorry for our negligence, and we have made correction. Please review line 416 in the section of plant materials and sampling.

  1. Materials and Methods. Plant materials and sampling. It is not clear how heartwood and sapwood were detected on the increment core. Did you make it only visually? How many growth rings were included in each tissue (outer sapwood, inner sapwood, transition zone, outer heartwood, inner heartwood)?

Response to comment:

Thank you for making essential comments. The yellow color of heartwood is deeper than sapwood that we can directly and accurately distinct heartwood and sapwood by visual observation in the field. In this study, 5-7 growth rings were included in outer sapwood, inner sapwood, outer heartwood, and inner heartwood. And, 2-3 growth rings were included in transition zone. We have detailed the method of wood sampling according to the reviewer’s comments. Please review line 417-422, and Figure 8B.

  1. Materials and Methods, line 423. «adial…» Maybe «Radial..»?

Response to comment:

We are very sorry for our negligence, and we have made correction. Please review line 430.

  1. Materials and Methods, line 238. «…2-amino-amino-3-(2-chloro-phenyl)-propionic acid…» It should be written in this way: «…2-amino-3-(2-chlorophenyl)-propionic acid…»

Response to comment:

We are very sorry for our negligence, and we have made correction. Please review line 445.

Special thanks to you for your good comments.

We tried our best to improve the manuscript and made some changes in the manuscript. These changes will not influence the content and framework of the paper. And here we did not list the changes but marked in red in revised paper.

We appreciate for Editors/Reviewers’ warm work earnestly, and hope that the correction will meet with approval.

Once again, thank you very much for your comments and suggestions.

Reviewer 2 Report

This work is a well designed and executed comprehensive metabolomic and transcriptomic analysis of the accumulating metabolites in the heart- and sapwood of the “golden-thread nanmu” tree of high economic value in China for its golden colored shades. The work identifies the metabolites providing the high value coloring for the wood. Analytical knowledge of the metabolites providing this benefit to the wood is expected to support breeding efforts for desired wood quality, and hence of high economic impact.

I only have some “cosmetic suggestions:

-     - In lines 36 and 412 when discussing the age of the trees samples were taken from, the authors use 60 and 80 a to denote age. It may be easier to read to the non-specialist reader if they would spell out “year” instead of “a” (“a” = annuum?) (or use “y”).

-         -  In Figure 2 title use “lipid” instead of "lipide”.

Author Response

List of Responses

Dear Editors and Reviewers:

Thank you for your letter and for the reviewers’ comments concerning our manuscript entitled “Integrative metabolomic and transcriptomic analysis reveals the mechanism of specific color formation in Phoebe zhennan heartwood (ijms-1909297)”. Thanks to the reviewer’s recognition of our work. And, those comments are all valuable and very helpful for revising and improving our paper, as well as the important guiding significance to our researches. We have studied comments carefully and have made correction which we hope meet with approval. Revised portion are marked in red in the paper.

Responds to the reviewer’s comments:

  1. In lines 36 and 412 when discussing the age of the trees samples were taken from, the authors use 60 and 80 a to denote age. It may be easier to read to the non-specialist reader if they would spell out “year” instead of “a” (“a” = annuum?) (or use “y”).

Response to comment:

We are very sorry for our negligence, and we have made correction. 60a and 80a were revised as 60 years old and 80 years old. Please review line 36 and 416.

  1. In Figure 2 title use “lipid” instead of "lipide”.

Response to comment:

We are very sorry for our negligence, and we have made correction. The word of “lipde” in the title of Figure 2 have been replaced by “lipid”. Please review the title in Figure 2.

Once again, thank you very much for your comments and suggestions.
